# Multi-Task Multimodal Fusion with Tabular Foundation Models for Pertussis Booster Response Prediction

## Abstract

Pertussis booster vaccination produces immune responses that vary widely across individuals in both peak magnitude and long-term durability, governed by partly distinct biological compartments. Most computational models target only one phase; jointly predicting both is non-trivial because the two endpoints reflect partly distinct biological processes, samples are small, and modalities have structured missingness. We propose a multi-task contrastive fusion architecture combining frozen TabPFN-v2 per-modality encoders, a dual-label supervised contrastive loss, modality dropout calibrated to empirical missingness, and missingness-masked attention fusion. On the CMI-PB pertussis booster subset (four modalities: antibody titers, cytokines, cell frequencies, gene expression; 44.9% of Task 1 subjects and 39.6% of Task 2 subjects missing at least one modality), we jointly predict peak response ($\log_2$(day 14 / day 0) IgG anti-pertussis-toxin fold change, n = 158) and durability ($\log_2$(day 120 / day 30) retention, n = 96). The model achieves test AUROC 0.797 (95% CI [0.621, 0.948]) for peak and 0.755 ([0.519, 0.945]) for durability, both significant under joint label permutation (p = 0.002, p = 0.045). Across logistic regression, XGBoost, and MLP baselines on raw features and TabPFN embeddings, the proposed model is the only method whose 95% CIs lie above chance on both tasks simultaneously.

## 1. Introduction

Pertussis booster vaccine responses vary widely across individuals, with peak antibody magnitude and long-term durability governed by partly distinct biological pro-

grams (Amanna et al., 2007; Gillard et al., 2024). Subjects who mount the strongest peak responses often exhibit the most rapid subsequent decline (the "boost-and-wane" trade-off), so a strong day-14 responder may be a poor day-120 retainer. Yet most computational models of vaccine response treat prediction as a single-task problem, typically targeting peak titer alone (Querec et al., 2009; Li et al., 2014), missing the clinical relevance of long-term protection.

The CMI-PB resource (Shinde et al., 2024) profiles pertussis booster responses across multiple immune modalities collected from 2020 to 2023. From this resource we use four (antibody titers, plasma cytokines, cell frequencies, gene expression) and face three challenges: (i) small sample size ($n = 158$ with a peak label, $n = 96$ also with a durability label), (ii) structured cohort-driven missingness (Fig. 1; 44.9% of Task 1 and 39.6% of Task 2 subjects missing at least one modality), and (iii) biological anti-correlation between the two prediction targets (Spearman $r = -0.58$). Tabular foundation models such as TabPFN (Hollmann et al., 2022; 2025) provide strong in-context representations on small tabular tasks, but their use in multimodal biomedical settings with heterogeneous missingness is largely unexplored.

We propose a multi-task contrastive multimodal fusion architecture that uses frozen TabPFN-v2 embeddings as per-modality encoders, aligned through a dual-label supervised contrastive loss and combined via missingness-masked attention fusion. Modality dropout calibrated to empirical missingness ensures robustness to arbitrary patterns of missing modalities at inference. The model jointly predicts peak (day 14 vs day 0 IgG-PT fold change) and durability (day 120 vs day 30 retention), achieves significant above chance performance on both tasks under label permutation, and recovers task-specific modality usage consistent with established vaccine immunology: cytokines drive peak prediction (Pulendran, 2014; Querec et al., 2009), antibody trajectories drive durability prediction (Amanna et al., 2007).

## 2. Method

### 2.1. Inputs and tasks

For each subject, we construct per-modality feature tables of baseline day 0 values and $\log_2$ fold changes at non label

[1]Anonymous Institution, Anonymous City, Anonymous Region, Anonymous Country. Correspondence to: Anonymous Author <anon.email@domain.com>.

Preliminary work. Under review by the International Conference on Machine Learning (ICML). Do not distribute.

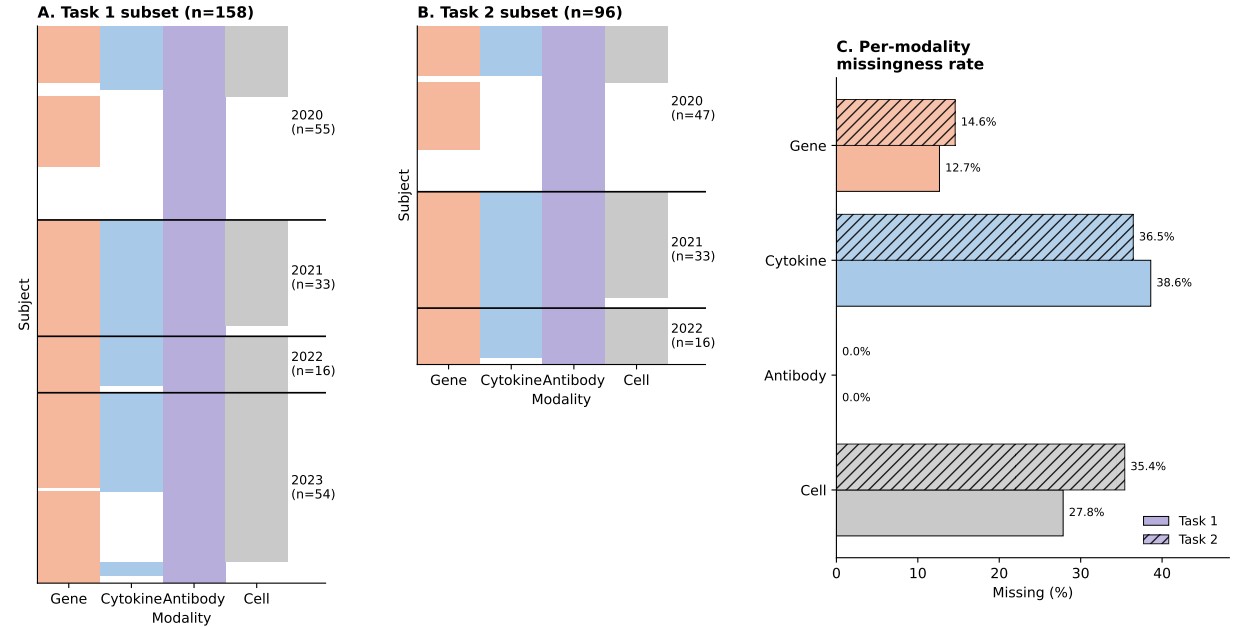

*Figure 1.* Cohort × modality missingness across modeled subjects. (A) Task 1 subset ($n = 158$); (B) Task 2 subset ($n = 96$); (C) per-modality missingness rate. Missingness is structural and cohort-driven, reflecting modality availability per cohort year rather than random subject dropout. This motivates an architecture that handles arbitrary modality subsets at both training and inference.

timepoints. Two binary targets are derived by median split: Task 1 (peak): $\log_2$(day 14 / day 0) IgG-PT fold change; Task 2 (durability): $\log_2$(day 120 / day 30) IgG-PT fold change. The two are anti-correlated (Spearman $r = -0.580$, $n = 96$). Pertussis-toxin antibody features are removed from inputs at all timepoints to prevent label leakage. Two binary metadata features (childhood priming, biological sex) are concatenated post-fusion. Full feature design and label contamination prevention are in Appendix B.

### 2.2. Architecture

The full architecture is shown in Fig. 2.

**Per-modality encoding.** Each modality's feature table is passed through a frozen TabPFN-v2 (Hollmann et al., 2025) encoder, returning a 1,536-dim in-context embedding per subject. TabPFN-v2 weights do not change during training. We use it as a frozen feature extractor, as our sample size is too small to train comparable encoders from scratch.

**Projection.** Independent two-layer MLP projection heads $g_m(\cdot)$ map each modality embedding to a shared 64-dim space (hidden 256), with $\ell_2$-normalization placing outputs on the unit sphere required by the contrastive loss.

**Dual-label supervised contrastive loss.** A SupCon (Khosla et al., 2020) objective pulls together embeddings of subjects forming a positive pair and pushes apart negatives. We

extend single-label SupCon to a dual-label variant: two subjects form a positive pair if they agree on the Task 1 *or* Task 2 label, so both tasks shape the geometry. Subjects with missing Task 2 labels participate via Task 1 alone (temperature $\tau = 0.3$, contrastive weight $\lambda = 0.1$). The OR construction is necessary because the two labels are anti-correlated; an AND construction would yield too few positives per batch.

**Modality dropout.** During training, each modality is independently zeroed with probability $p = 0.4$, with the constraint that at least one is retained. The per-batch probability of at least one dropped modality is $1-(1-p)^4 \approx 0.87$, deliberately exceeding the empirical inference-time missingness rate (44.9%). This ensures that any modality combination encountered at inference is in-distribution.

**Missingness-masked attention fusion.** A learned query $\boldsymbol{w} \in \mathbb{R}^{64}$ computes attention scores over present modalities; absent modalities (whether from real missingness or dropout) are excluded from the softmax denominator:

$$\alpha_m^{(i)} = \frac{\exp(\boldsymbol{w}^\top \boldsymbol{h}_m^{(i)})}{\sum_{m': \mu_{m'}^{(i)}=1} \exp(\boldsymbol{w}^\top \boldsymbol{h}_{m'}^{(i)})} \quad (1)$$

The fused vector is a convex combination of present modalities only, with no imputation. The same fusion code handles training time and inference time missingness without any mode switch.

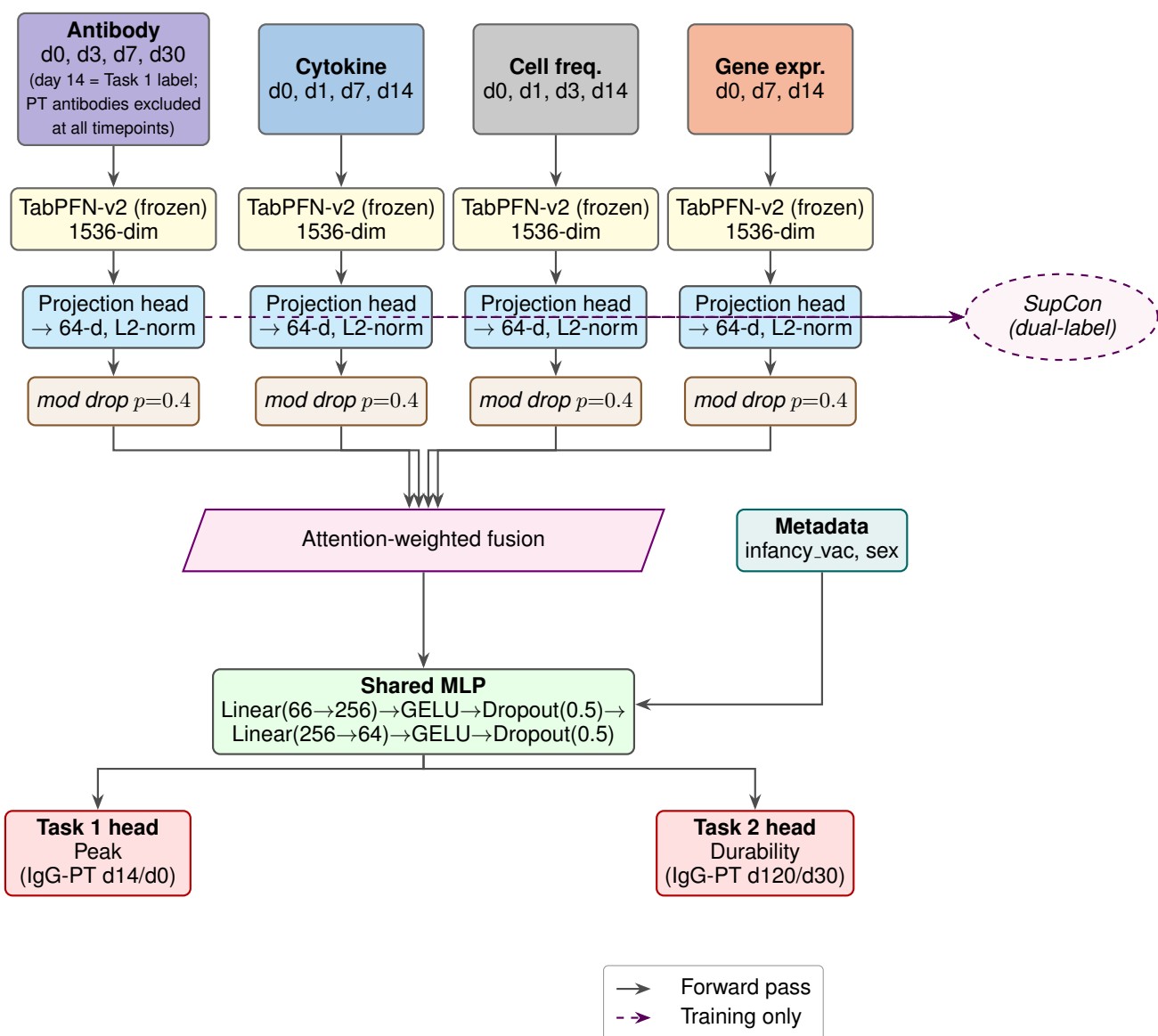

*Figure 2.* Multi-task contrastive multimodal fusion architecture. Four immune modalities are independently encoded by frozen TabPFN-v2 encoders and projected to a shared 64-dim space via per-modality MLP heads with $\ell_2$ normalization. A dual-label supervised contrastive loss (dashed arrows; training only) aligns same-class representations across modalities. Modality dropout ($p = 0.4$; training only) masks each modality independently. An attention-weighted fusion layer combines present modalities into a single representation, which is concatenated with subject metadata, passed through a shared MLP, and branched into two classification heads.

**Heads and loss.** The fused vector is concatenated with metadata, passed through a shared MLP, and branched into two linear heads (Task 1, Task 2). Total loss is

$$\mathcal{L} = \mathcal{L}_{\text{CE}}^{(\text{T1})} + w_{\text{T2}} \cdot \mathcal{L}_{\text{CE}}^{(\text{T2})} \cdot \mathbf{1}[y_{\text{T2}} \neq -1] + \lambda \cdot \mathcal{L}_{\text{con}} \quad (2)$$

with $w_{\text{T2}} = 2.0$ to compensate for masked-out subjects with missing durability labels. Full hyperparameters in Appendix C.

## 3. Experiments

### 3.1. Setup

We use a fixed stratified train/val/test split ($n_{\text{train}} = 94$, $n_{\text{val}} = 32$, $n_{\text{test}} = 32$ for Task 1; Task 2 test set $n = 21$). We report bootstrap 95% CIs ($B = 1000$) and joint label-permutation p-values ($N = 1000$ retraining runs). Baselines: logistic regression, XGBoost, and TabMLP, on (i) concatenated raw features with mean imputation, and (ii) concatenated TabPFN-v2 embeddings (6,144-dim).

*Table 1.* Architectural ablation. Each row turns one component off.

| Configuration | T1 AUROC | T2 AUROC |
|---|---|---|
| Full (preferred) | **0.797** | **0.755** |
| No contrastive | 0.566 | 0.682 |
| No modality dropout | 0.621 | 0.609 |
| Neither | 0.719 | 0.673 |
| No T2 up-weighting | 0.668 | 0.673 |

*Table 2.* Test AUROC: baselines vs ours. †: lower CI below 0.5; ‡: degenerate.

| Method | T1 | T2 |
|---|---|---|
| *Raw features (mean-imputed)* | | |
| LogReg | $0.699^{\dagger}$ | 0.791 |
| XGBoost | 0.781 | $0.645^{\dagger}$ |
| TabMLP | $0.594^{\dagger}$ | 0.777 |
| *TabPFN-v2 embeddings* | | |
| LogReg | **0.816** | $0.609^{\dagger}$ |
| XGBoost | $0.500^{\ddagger}$ | $0.364^{\dagger,\ddagger}$ |
| TabMLP | 0.785 | $0.500^{\ddagger}$ |
| Preferred (ours) | 0.797 | **0.755** |

## 3.2. Predictive performance

The full configuration achieves test AUROC 0.797 (95% CI $[0.621, 0.948]$) for Task 1 and 0.755 ($[0.519, 0.945]$) for Task 2. Joint label permutation yields $p = 0.002$ (Task 1, $\sim 3\sigma$ above null mean) and $p = 0.045$ (Task 2, $\sim 1.7\sigma$); both significant at $\alpha = 0.05$. Null distributions are well-calibrated near chance (Appendix D).

## 3.3. Per-modality contribution

Leave-one-out (LOO) and keep-one-out (KOO) analyses on the trained model (complete-case test subset) reveal task-specific modality usage. For Task 1, **cytokine** dominates: standalone KOO AUROC 0.935 (vs all-modalities 0.888) and largest LOO drop ($+0.068$). Cell frequency is the weakest standalone predictor (KOO 0.783) and removing it slightly *improves* the ensemble ($-0.058$ LOO), suggesting noise rather than signal. For Task 2, **antibody** alone matches the full-ensemble value (KOO 0.735); other modalities drop sharply in isolation (0.624–0.664). This task-specific usage emerges without supervision and is biologically coherent: peak response is driven by early innate cytokine signatures (Pulendran, 2014; Querec et al., 2009), while durability depends on long-lived plasma cell maintenance reflected in antibody trajectories (Amanna et al., 2007). Detailed plots in Appendix E.

## 3.4. Architectural ablation

Removing the contrastive loss drops Task 1 AUROC by 0.231, the largest single-component effect. Removing modality dropout drops it by 0.176. Notably, "Neither" (0.719) is *better* than either single-ablation configuration, indicating destructive interaction: contrastive alignment on full-modality data sharpens a poorly conditioned representation. The Full configuration is the only one above 0.79 on Task 1. The $w_{T2} = 1$ row shows that proper Task 2 weighting also improves Task 1 (a 0.129 drop when removed), evidence of multi-task transfer.

## 3.5. Comparison to baselines

On Task 1, the multi-task model and the strongest TabPFN-embedding baselines achieve comparable AUROC (LogReg 0.816, TabMLP 0.785, ours 0.797) with overlapping CIs: consistent with the contribution analysis showing peak is dominated by a single highly informative modality. On Task 2, every baseline either falls below chance or degenerates: durability requires integration across modalities and a label structure (the dual-label SupCon) that no concatenation baseline can access. The proposed multi-task architecture is the only method whose 95% lower CI bound is above chance on both tasks simultaneously: this joint reliability is its distinctive contribution. The raw-feature baselines also face a structural disadvantage: with substantial per-modality missingness, mean imputation fills entire blocks with training-set averages, while our attention fusion excludes absent modalities entirely.

## 4. Discussion and Limitations

The architecture recovers, without supervision, the textbook distinction between acute (cytokine-driven) and long-term (antibody-driven) humoral response phases, while delivering jointly above chance performance on two anti-correlated tasks under realistic missingness. The framework is naturally extensible to external cohorts and other vaccines where similar multi-omic profiling exists.

**Limitations.** Sample size is small ($n = 158 / 96$), producing wide CIs. Cross-cohort transportability is not directly assessed: training on 2020-2022 and testing on 2023 would be a natural test but is constrained by the absence of Task 2 labels for 2023. Per-modality contribution analyses describe the behavior of a single trained model and should not be read as universal biological claims.

## A. Dataset Details

We use the CMI-PB (Computational Models of Immunity: Pertussis Boost) dataset (Shinde et al., 2024), a longitudinal multi-omic resource for pertussis booster (Tdap) vaccination. We use four modalities: plasma antibody titers, plasma cytokine concentrations (Olink NPX), PBMC frequencies, and PBMC gene expression. All four were batch-corrected by the CMI-PB consortium using KNN imputation (Troyanskaya et al., 2001), median normalization within each dataset, and ComBat batch correction (Johnson et al., 2007). Subjects span four cohorts (2020–2023). We retain subjects with at least one modality and an available label, yielding $n = 158$ for Task 1 and $n = 96$ for Task 2 (subjects in common). The 2023 cohort lacks day-120 measurements and is excluded from Task 2 via masked loss. Missingness rates in the Task 1 subset: antibody 0%, gene 12.7%, cell 27.8%, cytokine 38.6%; 44.9% of subjects are missing at least one modality (Fig. 1 in main paper).

## B. Input Feature Design and Label-Leakage Controls

Input features are baseline day-0 values plus $\log_2$ fold changes at non-label timepoints (Table 3). Day-1 features are included for cytokine and cell, which capture innate-response dynamics that peak within 24-48 hours (Pulendran, 2014; Querec et al., 2009). Antibody and gene skip day 1; the most informative transcriptional signatures emerge at day 7 (Nakaya et al., 2011; Li et al., 2014). For linear-scale modalities (antibody, cell): $\mathrm{lfc}(f, t) = \log_2((f_t + 1)/(f_0 + 1))$. For log-scale modalities (cytokine NPX, gene expression): $\mathrm{lfc}(f, t) = f_t - f_0$.

*Table 3.* Per-modality experimental design. PT family = IgG-PT and its four subclasses, removed from antibody inputs at all timepoints.

| Modality | Input features | Timepoints | d1? |
|----------|----------------|------------|-----|
| Antibody | $x_0, \Delta_3, \Delta_7, \Delta_{30}$; PT family removed | d0, d3, d7, d30 | No |
| Cytokine | $x_0, \Delta_1, \Delta_7, \Delta_{14}$ | d0, d1, d7, d14 | Yes |
| Cell freq | $x_0, \Delta_1, \Delta_3, \Delta_{14}$ | d0, d1, d3, d14 | Yes |
| Gene expr | $x_0, \Delta_7, \Delta_{14}$ | d0, d7, d14 | No |

**Label-leakage controls.** Day-14 antibody (Task 1 numerator) and day-120 antibody (Task 2 numerator) never appear as inputs. PT-family antibodies are removed from inputs at every timepoint to avoid proximal leakage via within-subject autocorrelation. Anti-PT signal is still represented via the d3/d7 IgG titers of other pertussis antigens (FIM2/3, FHA, PRN), which are also acellular vaccine components and elicit correlated antibody responses (Edwards & Decker, 2018; Burdin et al., 2017). Gene-expression preprocessing (variance filtering to top 2,000 features, standardization) is fit on training data only.

## C. Hyperparameters

The projection heads, given a 1,536-dim TabPFN embedding $z$:

$$\boldsymbol{h}_m^{(i)} = \frac{g_m(\boldsymbol{z}_m^{(i)})}{\|g_m(\boldsymbol{z}_m^{(i)})\|_2}, \quad g_m(\boldsymbol{z}) = W_2^{(m)} \sigma\left(\mathrm{LN}(W_1^{(m)}\boldsymbol{z} + \boldsymbol{b}_1^{(m)})\right) + \boldsymbol{b}_2^{(m)} \tag{3}$$

with $\sigma$ = GELU, LN = LayerNorm. No weight sharing across modalities.

Optimization: AdamW (Loshchilov & Hutter, 2019), cosine annealing, gradient clipping at norm 1.0, early stopping on mean validation AUROC.

## D. Permutation Test and Bootstrap CIs

**Permutation test.** For each of $N = 1000$ permutations: shuffle Task 1 labels across all subjects and Task 2 labels independently across subjects with them; retrain the full pipeline on the shuffled dataset using the same train/val/test split; record test AUROC. The one-sided p-value is $p = \dfrac{1 + |\{i : \mathrm{null}_i \geq \mathrm{obs}\}|}{N + 1}$ (Ojala & Garriga, 2010). Joint shuffling is required because the dual-label SupCon would otherwise leak unshuffled-task signal.

**Bootstrap CIs.** For each of $B = 1000$ resamples: sample $n_{\mathrm{test}}$ test subjects with replacement, compute AUROC using the

*Table 4.* Hyperparameter configuration.

| Hyperparameter | Value |
|---|---|
| Projection dim / hidden | 64 / 256 |
| Shared MLP hidden dims | (256, 64) |
| Dropout | 0.5 |
| Contrastive weight ($\lambda$) / temperature ($\tau$) | 0.1 / 0.3 |
| Modality dropout ($p$) | 0.4 |
| Task 2 weight ($w_{\text{T2}}$) | 2.0 |
| Learning rate / weight decay | $10^{-2}$ / $10^{-3}$ |
| Batch size / max epochs / patience | 32 / 60 / 10 |

trained model's fixed predictions, discard single-class resamples, take the 2.5th and 97.5th percentiles. CIs reflect evaluation uncertainty given a single seed; multi-seed evaluation is deferred.

**Null distributions.** Task 1: mean 0.509, SD 0.098, observed 0.797 ($\sim 3\sigma$, $p = 0.002$). Task 2: mean 0.501, SD 0.147, observed 0.755 ($\sim 1.7\sigma$, $p = 0.045$).

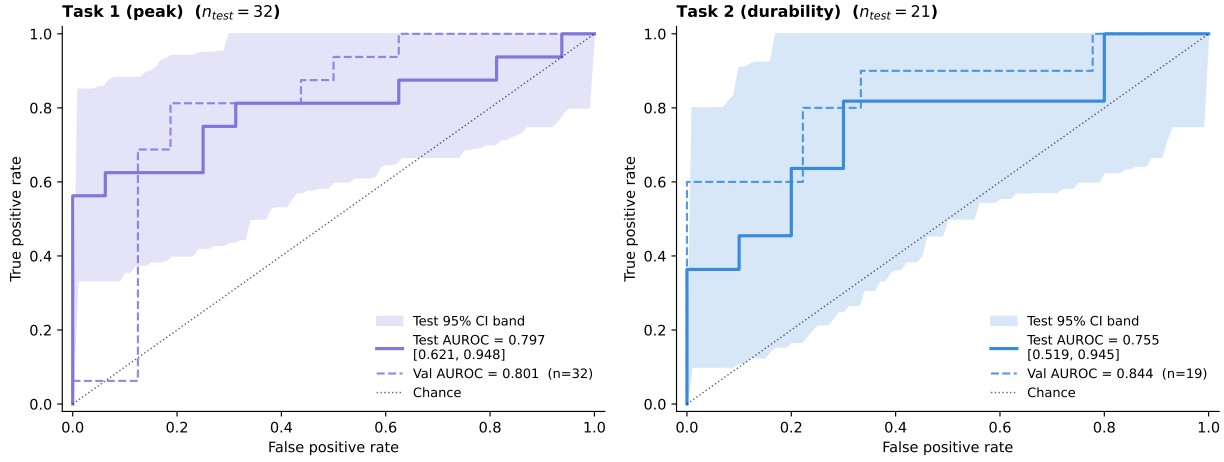

*Figure 3.* ROC curves on the held-out test set with 95% bootstrap confidence intervals. Left: Task 1 ($n_{\text{test}} = 32$), test AUROC 0.797, val 0.801. Right: Task 2 ($n_{\text{test}} = 21$), test AUROC 0.755, val 0.844.

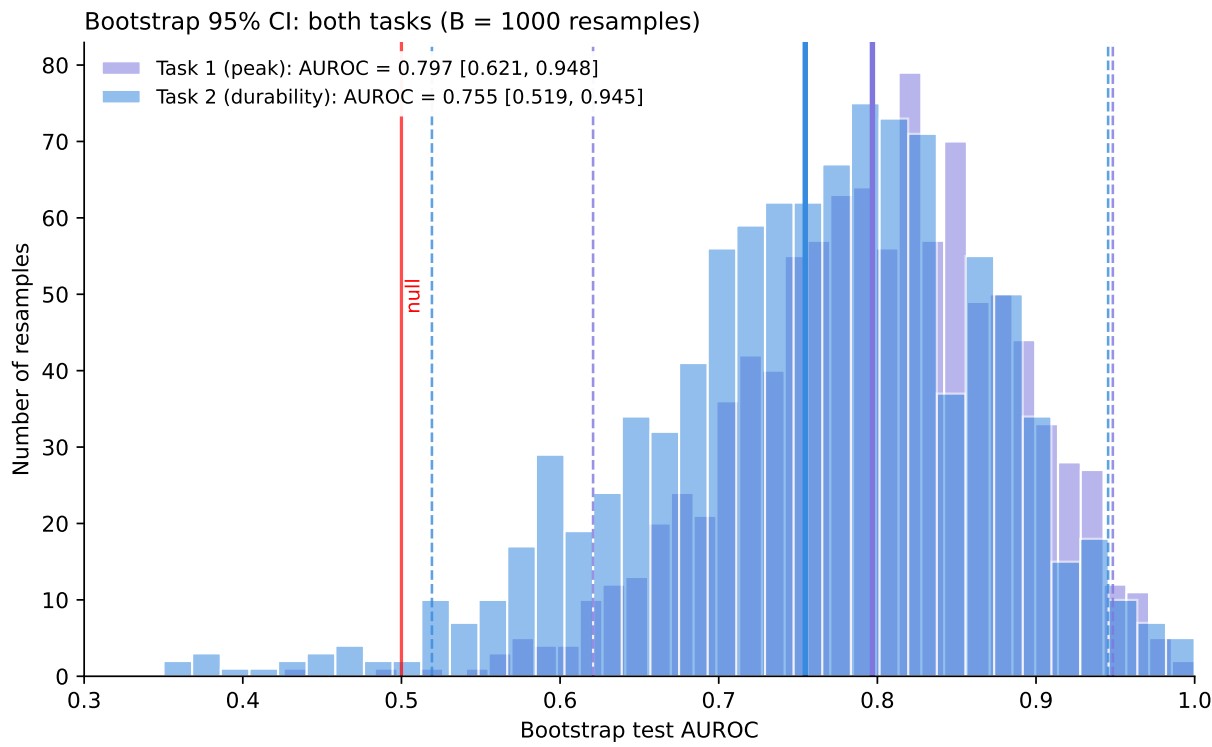

*Figure 4.* Bootstrap AUROC distributions over $B = 1000$ resamples. Solid lines: observed AUROC; dashed lines: 95% CI bounds; red: chance (0.5).

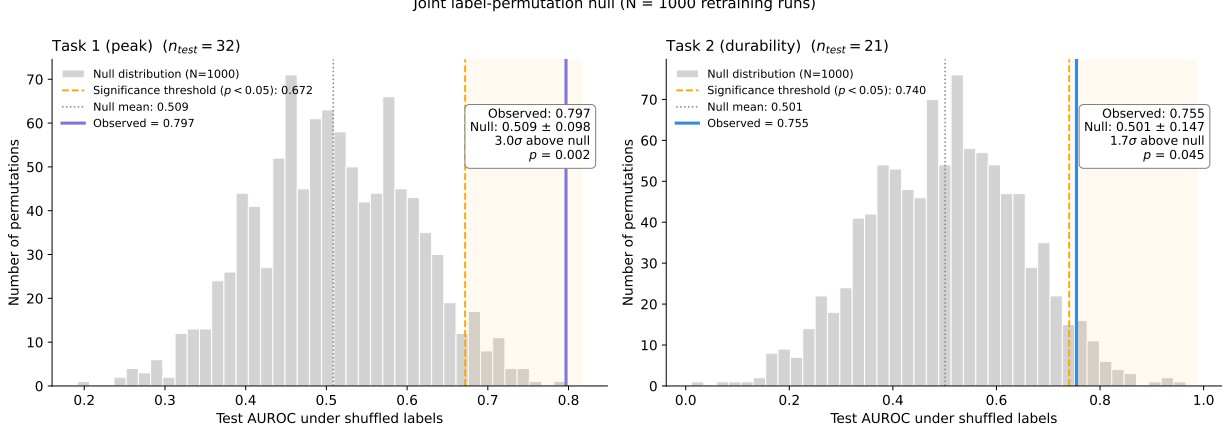

*Figure 5.* Joint label-permutation null distributions ($N = 1000$ retraining runs). Dashed orange: $p < 0.05$ threshold. Task 1: observed $0.797$, $p = 0.002$. Task 2: observed $0.755$, $p = 0.045$.

## E. Per-Modality Contribution and Graceful Degradation

**Leave-one-out (LOO):** mask one modality at inference; recompute test AUROC. **Keep-one-out (KOO):** retain only one modality at inference. Both on the trained preferred model, no retraining. Reference test AUROC on the complete-case subset is $0.888$ (Task 1) and $0.735$ (Task 2).

**Graceful degradation.** For each modality, mask a fraction $\rho \in \{0, 0.1, 0.2, 0.3, 0.5, 0.7, 1.0\}$ of test subjects (fixed seed). On Task 1, AUROC stays within $\sim 0.02$–$0.03$ of baseline across the full range of $\rho$ for every modality; on Task 2, performance stays close to baseline up to $\rho \approx 0.2$, with cell-frequency missingness the most damaging beyond that point.

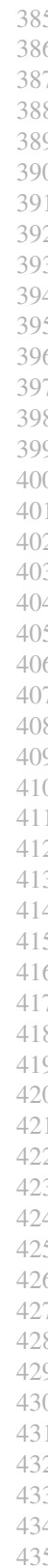

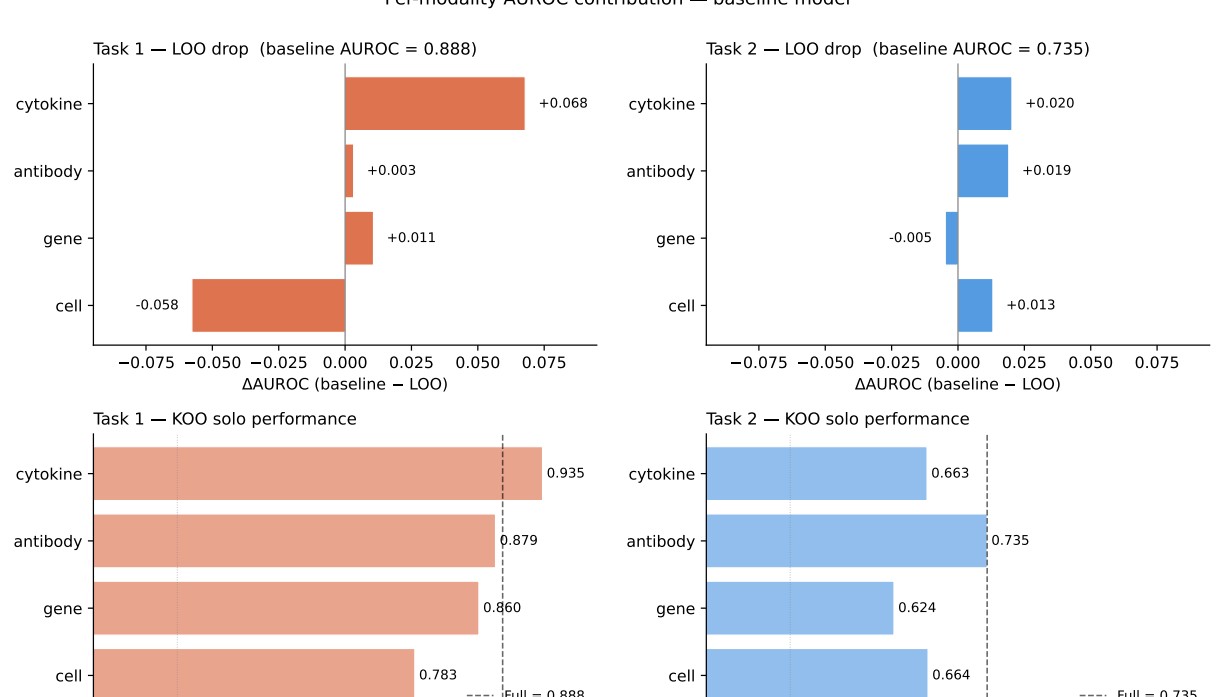

*Figure 6.* Per-modality contribution. Top: leave-one-out (LOO) drop, $\Delta =$ baseline $-$ LOO. Bottom: keep-one-out (KOO) standalone AUROC. Task 1: cytokine dominates (KOO 0.935, LOO +0.068); cell hurts the ensemble (LOO $-0.058$). Task 2: antibody alone matches the full ensemble (KOO $= 0.735$).

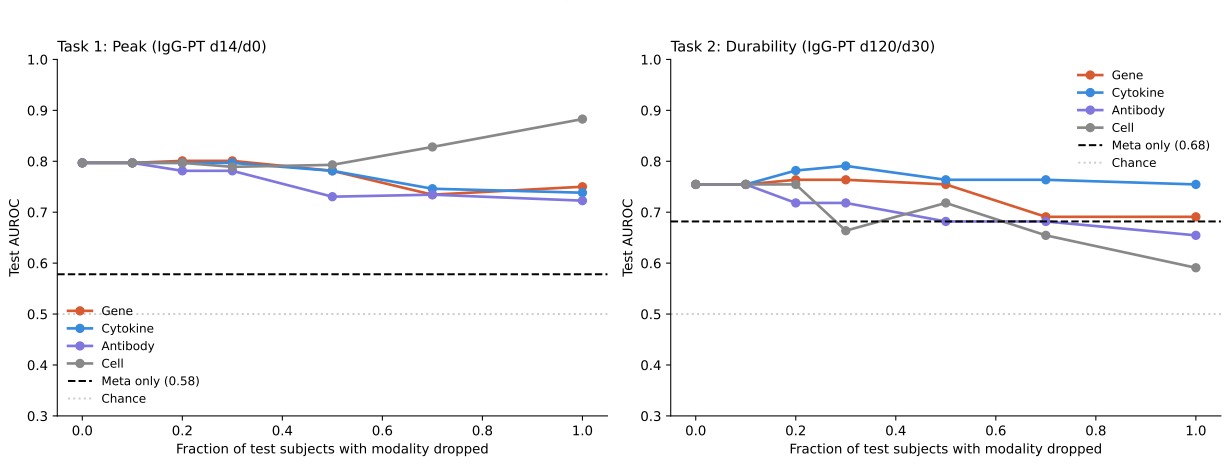

*Figure 7.* Graceful degradation under inference-time modality missingness. For each modality, a fraction $\rho$ of test subjects have that modality randomly masked at inference. Dashed black line: meta-only baseline.

## F. Anti-Correlation of Peak and Durability

The two binary tasks are statistically dissociated rather than redundant: Spearman $r = -0.580$ ($p = 3.97 \times 10^{-10}$, $n = 96$); Cohen's $\kappa = -0.520$. The $2 \times 2$ quadrant counts confirm asymmetry: 44 subjects fall in high peak / low durability, 30 in low peak / high durability, 20 in high peak / high durability, only 4 in low peak / low durability. This justifies the multi-task

440 framing.

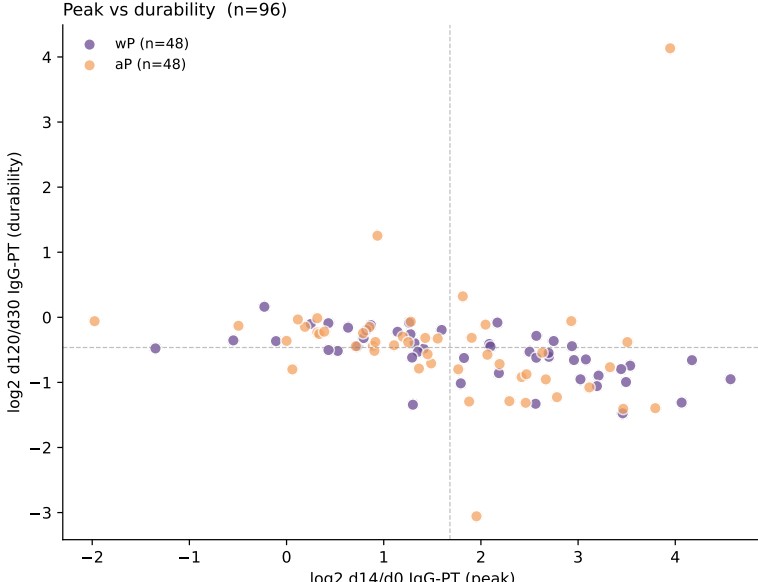

*Figure 8.* Peak vs durability scatter ($n = 96$). Spearman $r = -0.58$, Cohen's $\kappa = -0.52$. Dotted vertical: peak median (Task 1 binary cutoff). Dashed horizontal: durability median (Task 2 binary cutoff).

## G. Data and Code Availability

CMI-PB data are publicly available at `https://www.cmi-pb.org` (Shinde et al., 2024). Code to reproduce the analyses will be made available upon publication.

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
