# OpenReview forum: "Multi-Task Multimodal Fusion with Tabular Foundation Models for Pertussis Booster Response Prediction"
_ICML.cc/2026/Workshop/FMSD — FMSD @ ICML 2026 Poster_

### Official Review · Reviewer_T94d · 2026-05-18
**Reviewer finds this paper a decently formulated work with good secondary analysis. While the secondary analysis (modality LOO/KOO, and ablation) is good, initial analysis (specifically regarding model comparisons) could be improved. It is also not well justified and cannot stand on its predictive results alone to immediately merit its inclusion.**

**Rating:** 5
**Confidence:** 4

**Review:**

## Summary
&emsp; Authors present a model in which vaccine response for the pertussis booster where both immediate and delayed vaccine response are the predictive targets, representing the short and long-term protection induced by the vaccine. Authors use TabPVN-v2 as a frozen multi-modal encoder into an attention mechanism with a standard double prediction head MLP to predict both tasks. Comparison is then made to a small number of models for each task on raw and embedded features.

## Strengths
&emsp; The problem setting is well-formed and described, with a decent amount of secondary analysis of model performance. Including an exploration of the individual modality performance and contribution to the performance in the multi-modal setting and ablation of the architecture used. Specifically there is a good amount of model analysis in the supplemental materials.

## Areas Of Improvement
&emsp; I believe more appropriate models exist for comparison with the prediction tasks presented. Mostly the use of a regularized regression model instead of a vanilla logistic regression model. As with many clinical/biomedical applications the number of observations are low but the dimensionality is high. This is exactly the use case for a regularized regressor like LASSO, ridge-regression, or the Elastic-Net.

&emsp; While the analytical setup is good a description of how the authors arrived at their specific train/test/val split is necessary; Was it random split or was it selected via some other criteria. Additionally understanding the distributional characteristics of the two prediction tasks is needed to understand how out-of-distribution these sets are.

&emsp; Justification for why a joint predictive model is needed/beneficial should be made more explicit and discussed more. As it stands it appears that LogReg is superior in predictive performance (depending on features used). Authors state “The proposed multi-task architecture is the only method whos 95% lower CI bound is above change on both tasks simultaneously: this joint reliability is its distinctive contribution”, however I don’t know why I or anyone else should care about this characteristic. This reviewer however does believe that a joint model could have beneficial characteristics in a setting with anti-correlated prediction tasks. Specifically in terms of model explainability/interpretability. For instance a sparse auto-encoder applied to the attention mechanism may relay some interesting feature differences between these two tasks which could be biologically relevant. This is outside the scope of the paper but is the sort of justification I would want discussed.

## Detailed Comments
&emsp; There are a few clinically/biologically relevant factors relating to the task that should be discussed in the main text or supplemental. Pertussis is not described, a short description accessible to a broad audience to understand the medical context would help to contextualize this work. Similarly since the task is about the pertussis booster, and not the initial vaccine schedule, contextualization of the task in regards to this should be made. Does everyone get/need a booster? Does the fact that this is a booster induce a survivor bias?
Authors allude to previous models for pertussis vaccine predictions (lines 016-017) in the abstract. There however is no reference/discussion of these models. This would be the most direct comparison to their own work and should be included in some form.
I believe table 2 has the wrong T2 column elements bolded as the LogReg element has the best AUROC.

## Justification for Score
&emsp; Work authors present is a well structured paper constituting a significant amount of work, albeit with a fair amount of room for improvement. This reviewer believes that it could benefit from larger discussion with the community for its improvement and could be accepted. This is however dependent on the strength of competing submissions.

---

### Official Review · Reviewer_ZEKo · 2026-05-21
**Strong workshop submission on multimodal tabular foundation model fusion for vaccine response prediction under structured missingness**

**Rating:** 7
**Confidence:** 3

**Review:**

Review

This paper proposes a multi-task multimodal framework for predicting both peak and long-term durability of pertussis booster vaccine response using heterogeneous immune profiling data. The method combines frozen TabPFN-v2 modality-specific encoders with a dual-label supervised contrastive objective, modality dropout calibrated to empirical missingness, and missingness-masked attention fusion. Experiments on the CMI-PB pertussis booster dataset show statistically significant performance above chance for both prediction tasks simultaneously, outperforming several conventional baselines under structured multimodal missingness.

Overall, I found the paper well-motivated, technically sound, and highly relevant to the workshop theme. The work addresses a realistic structured-data setting involving small-sample multimodal biomedical prediction, which is an appropriate and compelling application area for tabular foundation models.

Strengths

1. Strong relevance to the workshop
    The paper is highly aligned with the goals of the Foundation Models for Structured Data workshop. In particular, it studies how frozen tabular foundation model embeddings can be integrated into multimodal biomedical prediction pipelines under realistic small-data conditions.
2. Well-motivated biological problem formulation
    Modeling both peak response and durability jointly is biologically meaningful and non-trivial. The observation that the two endpoints are anti-correlated (r=-0.58) provides a convincing justification for the proposed multi-task formulation instead of treating the problem as independent single-task prediction. The paper does a good job motivating why durability prediction is clinically important and underexplored.
3. Thoughtful handling of missing modalities
    The missingness-aware design is one of the strongest aspects of the paper. The use of modality dropout combined with masked attention fusion is elegant and practically reasonable for structured biomedical datasets where missingness is cohort-driven rather than random. I especially appreciated that the same fusion mechanism is used consistently during training and inference.
4. Careful experimental methodology
    The authors include:
    * bootstrap confidence intervals,
    * joint label permutation testing,
    * architectural ablations,
    * modality contribution analysis,
    * graceful degradation experiments.
    This level of evaluation rigor is stronger than many workshop submissions. The permutation-testing setup is particularly valuable given the small sample size.
5. Interesting empirical findings
    The observation that cytokines dominate peak prediction while antibody trajectories dominate durability prediction is biologically coherent and increases confidence that the model is learning meaningful structure rather than exploiting artifacts.

Weaknesses / Areas for Improvement

1. Limited sample size and wide uncertainty intervals
    The primary limitation is the relatively small cohort size (n=158 for Task 1 and n=96 for Task 2), which leads to wide confidence intervals, particularly for the durability task. For example, the Task 2 AUROC CI [0.519, 0.945] remains quite broad. Although the authors appropriately acknowledge this limitation, it still weakens confidence in robustness and reproducibility.
2. Novelty is somewhat incremental
    While the overall system is thoughtfully engineered, many components are individually established:
    * TabPFN embeddings,
    * supervised contrastive learning,
    * attention fusion,
    * modality dropout.
    The contribution is primarily in the integration of these techniques into a biomedical multimodal setting rather than introducing a fundamentally new modeling framework. This is acceptable for a workshop paper, but limits the methodological novelty.
3. Limited comparison to alternative multimodal fusion strategies
    The baselines focus mainly on standard tabular models using raw features or concatenated embeddings. It would strengthen the paper to compare against additional multimodal fusion approaches (e.g., gated fusion, cross-modal transformers, or late-fusion ensembles) to better isolate the contribution of the proposed architecture.
4. Single-split evaluation
    The experiments rely on a fixed train/validation/test split. Given the small dataset size, repeated cross-validation or multi-seed evaluation would provide a more stable estimate of generalization performance.
5. Frozen encoder design could be further justified
    The paper briefly argues that frozen TabPFN embeddings are preferable in the small-data regime, which is reasonable. However, additional discussion or ablation regarding fine-tuning versus frozen representations would improve the methodological justification.

Detailed Comments

* The paper is generally well-written and easy to follow. Figures 1 and 2 are particularly clear and effectively communicate both the missingness structure and overall architecture.
* The discussion around the OR-based dual-label SupCon formulation is insightful and biologically motivated.
* I appreciated the explicit discussion of label leakage prevention, especially removal of PT-family antibody features across all timepoints.
* The “graceful degradation” experiments are a nice addition and improve confidence in robustness under inference-time missingness.
* It may help readers if the authors provide more intuition regarding why contrastive alignment improves Task 1 so substantially relative to the no-contrastive ablation.
* The paper would benefit from a short discussion of computational cost and inference efficiency, especially since TabPFN-v2 embeddings are used for each modality independently.
* Some readers may question whether median-split binarization discards useful information from the continuous antibody trajectories. A brief discussion of this design choice would strengthen the framing.

Justification of Score

I view this as a solid workshop contribution with good empirical rigor, strong thematic relevance, and a realistic application setting. The work demonstrates that tabular foundation model embeddings can be effectively integrated into multimodal biomedical pipelines under severe missingness and limited data regimes.

The main limitations are the relatively incremental methodological novelty and the small dataset size, which makes it difficult to fully assess robustness and generalization. Nevertheless, the paper is carefully executed, statistically responsible, and likely to be interesting to the workshop audience.

Overall, I recommend acceptance.

---

### Official Review · Reviewer_7CUr · 2026-05-22
**Multi-Task Multimodal Fusion with Tabular Foundation Models for Pertussis Booster Response Prediction**

**Rating:** 6
**Confidence:** 3

**Review:**

## Summary

The paper proposes a multi-task multimodal fusion architecture that uses frozen TabPFN-v2 as per-modality encoders, combined via a dual-label supervised contrastive loss, modality dropout, and missingness-masked attention fusion, applied to pertussis booster response prediction on the CMI-PB dataset.

## Strengths

- Application-specific problem and solution: the dual-task framing (peak + durability) directly addresses the "boost-and-wane" trade-off in vaccine immunology, and the architectural choices (dual-label SupCon, missingness-masked attention, calibrated modality dropout) are tailored to the structural cohort-driven missingness in CMI-PB rather than being generic.
- Well-designed architecture and clear writing: each component (frozen TabPFN encoders, projection heads, dual-label contrastive loss, attention fusion) is motivated explicitly, the architectural ablation cleanly isolates each contribution, and the recovered task-specific modality usage (cytokine→peak, antibody→durability) provides a biologically coherent sanity check.

## Areas for Improvement

- Test sets are very small (n=32 for Task 1, n=21 for Task 2), reflected in extremely wide bootstrap CIs. The Task 2 lower bound is essentially at chance, making the headline result fragile.

- Other missingness-resilient multimodal architectures (FlexMoE [1], FuseMoE [2], MAESTRO [3]) have developed more principled mechanisms for handling missing modalities via mixture-of-experts routing and modality-specific gating. The paper would benefit from either comparing against these or discussing whether their ideas (MoE-style routing over present modalities, modality-aware gating) could complement or replace the current attention-fusion + modality-dropout design. That said, the small-n regime here understandably constrains end-to-end model development, so a simpler comparison against XGBoost (which handles missingness natively without imputation) would also be a useful and lightweight baseline to position the architectural complexity against.


**References**

[1] Yun et al. 2024. FlexMoE: Scaling Large-scale Sparse Pre-trained Model Training via Dynamic Device Placement.


[2] Han et al. 2024. FuseMoE: Mixture-of-Experts Transformers for Fleximodal Fusion.


[3] Mohapatra et al. 2025. MAESTRO : Adaptive Sparse Attention and Robust Learning for Multimodal Dynamic Time Series